# Characteristics and Outcomes of Colorectal Cancer Patients Cared for by the Multidisciplinary Team in the Reggio Emilia Province, Italy

**DOI:** 10.3390/cancers16132390

**Published:** 2024-06-28

**Authors:** Lucia Mangone, Francesco Marinelli, Isabella Bisceglia, Maria Barbara Braghiroli, Maria Banzi, Angela Damato, Veronica Iori, Carmine Pinto, Loredana Cerullo, Carlotta Pellegri, Maurizio Zizzo, Fortunato Morabito, Antonino Neri, Paolo Giorgi Rossi

**Affiliations:** 1Epidemiology Unit, Azienda USL-IRCCS di Reggio Emilia, 42123 Reggio Emilia, Italy; francesco.marinelli@ausl.re.it (F.M.); isabella.bisceglia@ausl.re.it (I.B.); mariabarbara.braghiroli@ausl.re.it (M.B.B.); paolo.giorgirossi@ausl.re.it (P.G.R.); 2Medical Oncology Unit, Azienda USL-IRCCS di Reggio Emilia, 42123 Reggio Emilia, Italyangela.damato@ausl.re.it (A.D.); carmine.pinto@ausl.re.it (C.P.); 3Unit of Gastroenterology and Digestive Endoscopy, Azienda USL-IRCCS di Reggio Emilia, 42123 Reggio Emilia, Italy; veronica.iori@ausl.re.it; 4Quality and Accreditation Office, Azienda USL-IRCCS di Reggio Emilia, 42123 Reggio Emilia, Italy; loredana.cerullo@ausl.re.it (L.C.); carlotta.pellegri@ausl.re.it (C.P.); 5Unit of Surgical Oncology, Azienda USL-IRCCS di Reggio Emilia, 42123 Reggio Emilia, Italy; maurizio.zizzo@ausl.re.it; 6Gruppo Amici Dell’Ematologia Foundation-GrADE, 42123 Reggio Emilia, Italy; fortunato.morabito@grade.it; 7Scientific Directorate, Azienda USL-IRCCS di Reggio Emilia, 42123 Reggio Emilia, Italy; antonino.neri@ausl.re.it

**Keywords:** colorectal cancer, stage, multidisciplinary team, recurrence, disease-free survival, death

## Abstract

**Simple Summary:**

Colorectal cancer remains a prevalent neoplasm affecting both genders. Despite advancements in screening techniques facilitating early detection, effective patient management remains paramount. This study investigates the impact of a multidisciplinary team (MDT) approach on patient outcomes. Results reveal a significant improvement in survival rates among MDT-followed patients compared to those lacking MDT oversight. Notably, no-MDT patients exhibited a twofold increase in mortality risk from both colon and rectal cancers. Furthermore, advanced age (>70 years) and advanced-stage disease (III and IV) emerged as pivotal risk factors. Consequently, prioritizing MDT intervention for these high-risk cohorts is imperative to optimize therapeutic strategies.

**Abstract:**

Colorectal cancer emerged as the third most prevalent malignancy worldwide, affecting nearly 2 million individuals in the year 2020. This study elucidates the pivotal role of a multidisciplinary team (MDT) in influencing the prognosis, as measured by relative survival rates, depending upon the stage and age. Cases recorded in an Italian Cancer Registry between 2017 and 2018 were included. Relative survival was reported at 1 and 3 years after diagnosis comparing MDT vs. no-MDT approaches. During the study period, 605 CRCs were recorded while 361 (59.7%) were taken care of by an MDT. Compared to no-MDT, MDT patients were younger with earlier stages and received more surgery. One year after diagnosis, survival was 78.7% (90% in MDT vs. 62% in no-MDT); stratifying by stage, in the MDT group there was no survival advantage for stage I (97.2% vs. 89.9%) and II (96.8% vs. 89.4%), but an advantage was observed for stage III (86.4% vs. 56.9%) and stage IV (63.7% vs. 27.4%). Similar values were observed at 3 years where a marked advantage was observed for stages III (69.9% vs. 35.1%) and IV (29.2% vs. 5.1%). The univariable analysis confirmed an excess risk in the no-MDT group (HR 2.6; 95% CI 2.0–3.3), also confirmed in the multivariable regression analysis (HR 2.0; 95% CI 1.5–2.5). Despite the increase in the number of MDT patients in 2018 (from 50% to 69%), this does not correspond to an improvement in outcome.

## 1. Introduction

Colorectal cancer (CRC) is the second most frequent neoplasm in Italy, with 43,700 new cases diagnosed each year and over 19,000 deaths per year [1]. Excessive consumption of red meat and sausages, refined flour and sugars, overweight and reduced physical activity, smoking, and excess alcohol are well-known risk factors [2,3,4]. Familial adenomatous polyposis (FAP) and Lynch syndrome have hereditary susceptibilities (2–5%) attributable to genetic mutations [5,6].

The 5-year survival rate in both males and females is 65%, and it has risen over time as a result of improved screening [7,8,9], therapeutic advanced approaches [10,11,12], and the development of the multidisciplinary team (MDT) [13,14,15]. The implementation of national screening for the detection using a faecal occult blood test, combined with the breakthroughs in treatment, has resulted in a significant improvement in cure rates and survival in this neoplastic disease [16,17]. In Italy, the implementation of screening programs in the early 2000s had an effect on mortality and incidence that has been observed in most programs [18,19,20]. According to Italian organizational recommendations, the implementation of screening programs entailed the definition of diagnostic and therapeutic pathways, with the local level favouring the link between outpatient and inpatient care, laboratory, endoscopy, surgery, oncology, and pathology services, thus creating a favourable environment for the establishment of MDT [21]. MDT promotes collegial discussion of a patient’s case, to assist in developing a diagnostic strategy, validating a diagnosis, or deciding on treatment modalities [22]. It may increase the appropriateness of surgery [23] and therapy [24]. According to the literature, participating in MDT is linked to improved patient survival, especially in those with advanced colorectal cancer [25,26,27,28]. The number of patients receiving MDT care has risen steadily over the years. In France, the proportion of patients benefiting from MTD consultations increased from 32% in 2000 to 82% in 2018 [29,30]. Even when an MDT is active in a reference cancer centre, not all CRC occurring in a specific area or even present at the centre, are assessed and cared for by the MDT. The likelihood of not being followed by an MDT appears to be more frequently associated with advanced age, the presence of comorbidities, and the presence of less advanced tumours [22].

The initiation of CRC screening in Reggio Emilia in 2005 [31], heralded a significant reduction in both mortality and incidence rates attributable to high participation levels [32]. This implementation of CRC screening facilitated the development of comprehensive outpatient and inpatient protocols dedicated to the diagnosis and management of screen-detected cancers. In 2017, a comprehensive review of the diagnostic and therapeutic pathway for hospital management was conducted with the goal of including all eligible patients. This pathway was designed to include an MDT assessment and facilitate shared treatment decision-making among healthcare professionals. By integrating MDT evaluation and collaborative decision-making into the management process, the revised pathway aimed to optimize patient care and outcomes through a coordinated and multidimensional approach.

In this study, we described the differences between MDT and no-MDT CRC cases, comparing patient characteristics and survival outcomes.

## 2. Materials and Methods

### 2.1. Setting and Data Sources

The Reggio Emilia Cancer Registry (RE-CR), established in 1996, serves a population of 532,000 inhabitants. This registry systematically collects and maintains up-to-date data, extending its incidence data up to the end of 2021. Notably, the registry boasts a remarkable accuracy rate, with a microscopic confirmation rate as high as 93.2% for CRC cases and a low percentage of death certificate only (DCO) accounting for less than 0.1% [33]. This data collection methodology ensures the reliability and credibility of the information used in the study. The CR collects data and information following current flows to produce incidence, mortality, prevalence, and survival statistics for the resident population and demographic subgroups as required by the epidemiological report, defined by Law no. 29 of 22 March 2019 that regulates the CRs in Italy. This law exempts the registries from collecting informed consent. The procedures for conducting epidemiological analyses of the RE-CR data were approved by the provincial Ethics Committee of Reggio Emilia (Protocol no. 2014/0019740 of 4 August 2014).

### 2.2. Data Collection

The leading information sources of the RE-CR are pathologist reports, hospital discharge records, and mortality data, integrated with laboratory tests, diagnostic reports, and information from general practitioners. The work includes all 605 cases of colorectal cancer diagnosed in the province of Reggio Emilia in the period 2017–2018, without any selection between operated cases, urban residents, stage, etc. The cohort of 605 patients was sent to the MDT manager who added the variable MDT. Consequently, the cases were divided into two cohorts: 361 cases in the MDT group and 244 cases in the no-MDT group. All cases were defined based on the International Classification of Diseases for Oncology, Third Edition (ICD-O-3) as topography C18-C19 [34]. Information on stage (TNM 8th edition) [35], surgery, and chemotherapy was collected by consulting for each case the medical records in the hospital. Basic yes/no information is provided about chemotherapy and surgery, but it is unclear if these treatments are being used for palliative or curative purposes.

### 2.3. Description of the MDT Composition and Functions

All the diagnostic and therapeutic protocols for the major cancer sites were reviewed and updated in 2017. The systematic approach implemented the establishment of evidence-based decision-making protocols, delineation of functions, responsibilities, and interfaces among various healthcare services involved in patient care delivery, and consistent consultation with a cancer site-specific MDT to deliberate on the primary steps of the care pathways. This structured framework aims to ensure the integration of best practices, streamline communication channels between healthcare providers, and promote a cohesive and patient-centered approach to cancer management. Process and outcome quality indicators are used to monitor the entire process. Annual feedback on indicators is provided to all healthcare operators involved in the care. New protocols and MDT have been gradually implemented, starting with breast cancer in 2011 and CRC in 2017. The team consists of medical oncologists, radiologists, radiation oncologists, gastroenterologists and digestive endoscopists, surgeons, pathologists, and others who meet weekly to determine the best therapeutic solution for each patient based on age, social status, and the stage and characteristics of the tumour.

### 2.4. Statistical Analysis

The descriptive statistics by age at diagnosis (divided into 3 age groups: <50, 50–69, and over 69 years), method of diagnosis, stage, surgery, and chemotherapy stratified by group status (MDT vs. no-MDT) were calculated. The 1-year and 3-year relative survival of CRC registered in the period 2017–2018 was calculated using the Pohar Perme method. Relative survival is an estimate of net survival representing cancer survival in the absence of other causes of death. It is defined as the ratio of the proportion of observed survivors in a cohort of cancer patients to the proportion of expected survivors in a comparable set of cancer-free individuals. A multivariable Cox proportional hazard regression model was constructed to investigate the association between MDT and overall survival with time expressed in years. To account for potential confounding factors, this model was adjusted for covariates including age, disease stage, and tumour site. Additionally, another Cox regression model was employed to assess the association between year and OS, adjusted for the same covariates of age, stage, site, and surgery.

Furthermore, three Cox regression models were conducted encompassing the entire cohort, comprising both MDT cases and no-MDT cases. These modes were adjusted for multiple covariates, including age (categorized as <70 years and 70+ years), year of diagnosis (2017 and 2018), tumour site (colon and rectum), gender, and disease stage (I, II, III, IV). The time-to-event analysis (OS) by year and MDT adjusted for age, was performed using the Kaplan–Meier method. Analyses were performed using STATA 16.1 software. Given the descriptive nature of this study, we did not perform any formal statistical test of hypothesis and we did not fix any pre-defined threshold of significance. The reported 95% confidence intervals should be interpreted only as a measure of the precision of the point estimate; similarly, *p*-values should be interpreted as continuous variables reflecting the probability of observing a difference between two groups if the two groups were random samples of the same population, while we know that they are not.

## 3. Results

### 3.1. Characterization of the Patient Group by Age, Stage, Treatment, and MDT Approach

During the study period, 605 CRCs were recorded. Among these cases, 62.8% were diagnosed in patients aged 70 years and above. Regarding cancer stage distribution, 24.5%, 28.4%, 27.1%, and 17.9% of cases were categorized as stages I, II, III, IV, respectively. The distribution across the years of diagnosis revealed that 52.7% of cases occurred in 2017, while 47.3% were recorded in 2028. In terms of tumour site, 72.9% of CRC cases originated in the colon while 27.1% were located in the rectum. Notably, 361 patients (59.7%) received care facilitated by an MDT. Surgical therapy was undergone by 63% of patients, while chemotherapy was administered to 30.9% of cases as part of their treatment regimen (Table 1).

The 361 MDT patients exhibited distinct characteristics compared to the no-MDT patients. Specifically, MDT patients were younger and demonstrated an increase in management rates, rising from 45.4% in 2017 to 54.6% in 2018. Moreover, MDT patients displayed an enhanced morphological characterization of their tumours, were diagnosed at earlier stages, and were more likely to undergo major surgical interventions (Table 2).

### 3.2. Relative Survival Rates at 1 Year and 3 Years in MDT and no-MDT Patients, Stratified by Stage

One year after diagnosis, the OS rate was 78.7%, with higher survival rates in MDT (90%) compared to those without MDT involvement (62%). Stratifying by stage, no survival advantage for MDT patients in stage I (97.2% vs. 89.9%) and II (96.8% vs. 89.4%) was demonstrated, whereas significant differences were evident for stage III (86.4% vs. 56.9%) and stage IV (63.7% vs. 27.4%). At 3 years, the OS rate decreased to 65.2% with MDT patients exhibiting a substantially higher survival rate (78.5%) compared to their MDT counterparts (45.5%). No significant survival advantages were observed in stage I (92.6% vs. 83.6%) and II (93.6% vs. 74%), while significant disparities were evident for stages III (69.9% vs. 35.1%) and IV (29.2% vs. 5.1%).

Breaking down the data by T and N parameters, differences were more evident for advanced cancers: for T3–T4 tumours, the survival rates at 1 year was 87.9% vs. 71.5%, and at 3 years, 74.8% vs. 49.8%, favouring MDT patients. Similarly, for N+ cases the survival rates at 1 year were 85.4% vs. 63.2%, and at 3 year 61.5% and 32.1%, again demonstrating the survival advantage for MDT patients (Table 3).

### 3.3. Overall Survival and Multivariable Analysis

The univariable analysis shows a double associated-risk in patients no-MDT (HR = 2.6, 95% CI 2.0–3.3), adjusting for all available confounders (age, stage, site and surgery), the risk remains double even if slightly lower in the multivariate group (HR 2.0, 95% CI 1.5–2.5), while no risk is associated with the year of diagnosis (Table 4).

In the MDT group only (Appendix A), advanced age (>70 years) (HR 3.0; 05%CI 1.9–4.7), diagnosis in the year 2018 (1.4; 95% CI 1.0–2.1), and disease stage III (HR 3, 95% CI 1.7–5.3) and IV (HR 9.3 95% CI 5.2–16.8) were identified as variables significantly associated with an increased-associated risk of adverse outcomes. Similarly, these increased risks were observed within the no-MDT group as well. Finally, patients diagnosed in the years 2017–2018 being treated by MDT were associated with increased survival (Figure 1A) despite slightly higher survival rates observed in 2017 compared to 2018 (Figure 1B). 

## 4. Discussion

The aim of this study was to evaluate whether patients diagnosed with CRC and managed by an MDT exhibited superior survival outcomes and reduced risk of mortality compared to those without MDT involvement. The study encompassed all CRC diagnosed in the province of Reggio Emilia during the period 2017–2018, ensuring an unbiased and inclusive analysis without exclusion criteria.

The investigation was conducted in a province located in northern Italy recognized for its high incidence of tumours [36] but also commendable adherence to oncological screening programs which favored a progressive reduction in mortality rates and contributed to a decrease in cancer incidence, as documented by Campari et al. [31,32]. In our institution, the management of CRC is centralized within an oncological network, assuring uniform and standardized care delivery to patients across the entire province, irrespective of the initial hospital they accessed.

### 4.1. Characteristics of Patients in MDT and no-MDT

Our results are partially consistent with those of a previous French study. Reboux and colleagues [22] found that, during the study period, 20.5% of patients diagnosed with CRC were not presented in MDT meetings and were associated with ECOG-PS of 2 (OR 0.5, 95% CI 0.3–0.8), best supportive care (OR 0.05, 95% CI 0.0–0.4), and early death (OR 0.1, 95% CI 0.4–0.2). These results are consistent with our finding of a lower probability of being assessed by the MDT for patients with metastases, older age, and not receiving surgery or chemotherapy. On the contrary, the French study found also that patients with symptomatic tumours were more likely to be presented in MDT meetings than patients participating in mass screening (OR 2.26, 95% CI 1.19–4.3); stage I tumours were associated with non-presentation, probably because adjuvant treatment is not required. These findings are partially different from our experience where stage I and screening-age cancers are those with the highest probability of being included in MDT assessment and the difference is probably due to different organization of screening programs in the two health systems.

As regards the advanced stage, a population-based study showed that patients with stage IV were less often assessed by an MDT and less often selected for surgery, especially if compared with rectal patients [27]. In our study instead, we found no significant differences between patients with colon and rectal cancer, not even for advanced stages. However, age per se is not a reason for not discussing patients in an MDT setting. There is evidence today that elderly patients can receive the same benefit from chemotherapy as younger ones without a significant increase in toxicity [37,38].

### 4.2. Trying to Interpret the Association between MDT Assessment and Survival

Several studies showed an association between offering an MDT approach to patients with CRC and improved survival outcomes, particularly evident in cases of more advanced tumours [13,15]. Consistent with these findings, our study revealed the most relevant differences at 1 and 3 years, and only in stages III and IV. These findings underscore the pivotal role of MDT collaboration in optimizing patient management and improving prognosis, particularly in the context of more advanced CRC cases. This result is relevant if we consider that over 50% of CRC are diagnosed at an advanced stage and therefore an improvement in the multidisciplinary management of these patients from the beginning could lead to better outcomes. Similar results were also observed in ovarian cancer [39] where, in the MDT group, only stage III and IV patients had advantages in terms of mortality and disease-free survival compared to no-MDT. These consistent findings across different cancer types underline the importance of MDT involvement in cases of advanced-stage malignancies where coordinated and comprehensive care approaches are paramount.

It is important to underline that the composition and roles within MDT differ from one country to another [15]. Notably, key figures in these teams include radiologists and surgeons, especially in the context of primary CRC resection or any preoperative interventions, including cytoreduction procedures [40]. Their expertise and involvement play a pivotal role in ensuring comprehensive and effective care for CRC patients.

Despite differences in the compositions and functions, studies reporting the impact of MDT on survival are consistent. A Sweden study shows locally advanced CRC survival in the MDT group was significantly higher (80%) than in the non-MDT group (68%) both at 3 year and 5 year (73% and 60%, respectively) [41], similar results to those observed in our study. The study conducted by Hsu et al. [42], involving 25,766 patients with stage I–IV CRC, showed a significant association between MDT assessment and decreased mortality (HR 0.9, 95% CI 0.8–0.9). Similarly, a Scottish study by Munro et al. of 586 patients with CRC across all stages demonstrated notable differences in 5-year cause-specific survival rates. Patients who underwent MDT assessment exhibited a cause-specific 5-year survival proportion of 63.1% whereas those without had a markedly lower survival rate of 48.2% [26]. These results emphasize the positive impact of MDT assessment even on long-term survival outcomes for CRC patients.

Indeed, also in other studies one possible explanation for the association between MDT assessment and improved survival rates in CRC could be the selection of patients at better prognosis. It is plausible that individuals with a poorer prognosis, characterized by rapidly advancing disease or the absence of viable therapeutic options, might have a reduced likelihood of being included in MDT discussions. In such cases, the limited treatment options due to the aggressive nature of the disease could contribute to this exclusion, thereby influencing the observed association between MDT assessment and decreased mortality rates. In fact, the implementation of an MDT requires time so that older patients with more advanced stages are also included, as demonstrated in a recent study conducted on endometrial cancers [43]. In that study, it was shown that in the period 2013–2015 compared to 2016–2020 of the MDT activity, there was a shift in the selection towards elderly women, with advanced stages and residing in rural areas.

Although early deaths could be responsible also for some differences between MDT and no-MDT in our study, the two curves diverge throughout the entire follow-up and not just in the first months suggesting that factors beyond early deaths are at play.

In addition, older age and metastases at diagnosis are more present in patients not included in MDT, which also indicates potential disparities in healthcare organization and inclusion criteria within MDT meetings. This diversity in inclusion criteria might contribute to the observed inverse association between MDT assessment and mortality. In this landscape, the concept of reverse causality, where the presence of advanced disease influences MDT inclusion, remains a plausible explanation.

Factors unrelated to physician choice, such as socio-economic determinants and other non-clinical patient-related characteristics that were not measured, might introduce confounding biases that could impact the observed associations. A final consideration concerns the fact that in our province there was a sharp decline in the incidence of colorectal cancers from 2000 to 2018: the cases went from 394 to 286 (unpublished work). At the same time, metastatic tumours decreased from 90 (24%) to 50 (17%). For metastatic cases the median survival is 7 months, as reported in a previous work [44]: in the present study the median has just increased (8 months) but even in this case a better median value persists in the MDT group (15 months) vs. no-MDT (5 months). Screen-detected cancers were shown to have better prognosis even when stratified by stage [45], more likely due to residual lead time bias. In our study, screen-detected cancers have a higher probability of being included in MDT, as shown by the stage and age distribution. Nevertheless, better survival of screen-detected cancers cannot explain all the difference in survival since this is appreciable also in patients over 70, i.e., out of the screening target age, and in stage IV cancers, that are rarely screen-detected [46].

Even if our study could not demonstrate a causal association, there are also plausible mechanisms that could explain the association between the MDT discussion and better survival that are causal. Therefore, today the MDT discussion has become a standard, and it is considered increasingly important with the development of new treatment strategies. In a recently published study, a reduced rate of positive margins at resection was reported in CRC patients when MRI was discussed preoperatively by an MDT [47]. In this population-based study, resection of metastases was associated with improved survival: 37% 5-year survival vs. 2% 5-year survival in patients who did not have surgery for metastases. Finally, the MDT approach has been shown to increase adherence to clinical guidelines and overall improvement in decision-making processes even if other factors should be evaluated, including the time consumption for clinicians, thus increasing resource consumption and costs [48].

The strengths of this study include the utilization of population-based data, ensuring minimal selection bias. Comprehensive data collection efforts by registrars were pivotal, capturing details such as stage and treatment, which are typically not included in standard international cancer registries that primarily document tumour location and morphology. The study covers recent years, coinciding with the initial implementation of the MDT.

The limitations of the study encompass the absence of comorbidity, known to significantly influence patient outcomes. Additionally, detailed specifics on surgical interventions (curative vs. palliative) and the specific chemotherapy regimens administered were not fully documented.

## 5. Conclusions

Patients discussed by the MDT had better prognostic factors, were younger, and received more frequent surgery and chemotherapy. Notably, advanced CRC patients (stages III and IV) showed a better outcome if included in an MDT care pathway. Residual confounding and other uncontrolled factors could be other plausible explanations for the better survival of patients cared for by MDT.

## Figures and Tables

**Figure 1 cancers-16-02390-f001:**
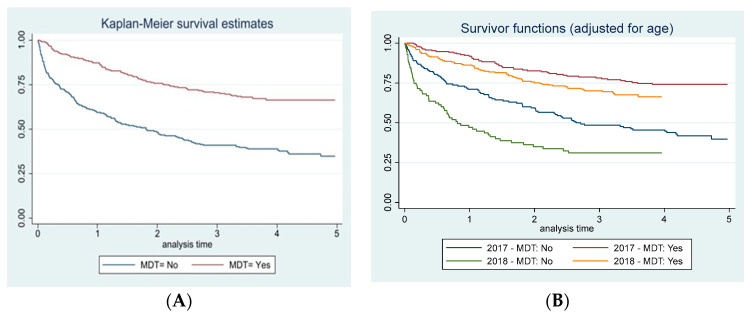
Reggio Emilia Cancer Registry, years 2017–2018. Kaplan–Maier curve of overall survival by MDT (**A**) and year of diagnosis (**B**) adjusted for age.

**Table 1 cancers-16-02390-t001:** Reggio Emilia Cancer Registry, years 2017–2018. Characteristics of patients.

Overall	605	
Age at diagnosis		
<50	32	5.3
50–69	193	31.9
70+	380	62.8
Stage		
I	148	24.5
II	172	28.4
III	164	27.1
IV	108	17.9
Unknown	13	2.1
Years		
2017	319	52.7
2018	286	47.3
Site		
Colon	441	72.9
Rectum	164	27.1
MDT		
Yes	361	59.7
No	244	40.3
Surgery		
Yes	381	63.0
No	179	29.6
Unknown	45	7.4
Chemotherapy		
Yes	187	30.9
No	373	61.7
Unknown	45	7.4

**Table 2 cancers-16-02390-t002:** Reggio Emilia Cancer Registry, years 2017–2018. Distribution of cases by age, year of diagnosis, stage, T, N, treatment, and MDT approach.

	MDT			
	Yes (n = 361)		No (n = 244)	
	*n*	%	*n*	%
Age at diagnosis				
<50	23	6.4	9	3.7
50–69	123	34.1	70	28.7
70+	215	59.5	165	67.6
Year of diagnosis				
2017	164	45.4	155	63.5
2018	197	54.6	89	36.5
Method of diagnosis				
Histological	360	99.7	221	90.6
Clinical/instrumental	1	0.3	23	9.4
Stage				
I	111	30.8	37	15.2
II	102	28.3	70	28.7
III	103	28.5	61	25.0
IV	44	12.1	64	26.2
Unknown	1	0.3	12	4.9
T				
T1	76	21.1	23	9.4
T2	52	14.4	14	5.7
T3	156	43.2	78	32.0
T4	60	16.6	55	22.5
Unknown	17	4.7	74	30.3
N				
N0	206	57.1	92	37.7
N1	69	19.1	34	13.9
N2	40	11.1	13	5.3
Unknown	46	12.7	105	43.0
Surgery				
Yes	263	72.8	118	48.4
No	70	19.4	109	44.7
Unknown	28	7.8	17	6.9
Chemotherapy				
Yes	121	33.5	66	27.0
No	212	58.7	161	66.0
Unknown	28	7.8	17	7.0

**Table 3 cancers-16-02390-t003:** Reggio Emilia Cancer Registry, years 2017–2018. 1-year and 3-year relative survival in the entire cohort and after categorizing by stage and MDT approach.

	1-Year						3-Year					
	MDT Yes		MDT No		Total		MDT Yes		MDT No		Total	
	%	95% CI	%	95% CI	%	95% CI	%	95% CI	%	95% CI	%	95% CI
Entire cohort	90.0	85.7–93.0	62.0	55.3–68.0	78.7	74.9–82.0	78.5	72.5–83.4	45.5	38.1–52.5	65.2	60.4–69.6
Stage												
I	97.2	86.9–99.4	89.9	70.0–96.9	95.4	88.1–98.2	92.6	79.5–97.5	83.6	57.0–94.4	90.4	79.9–95.6
II	96.8	84.5–99.4	89.4	76.6–95.4	93.8	86.9–97.1	93.6	75.6–98.5	74.0	57.7–84.7	85.6	75.7–91.7
III	86.4	76.8–92.2	56.9	43.0–68.7	75.5	67.4–81.9	69.9	57.8–79.2	35.1	21.4–49.0	57.1	47.5–65.5
IV	63.7	47.3–76.1	27.4	18.6–36.8	42.1	33.3–50.7	29.2	16.0–43.7	5.1	1.4–11.5	14.7	8.7–22.1
T												
T1–T2	97.8	87.9–99.6	92.2	71.9–98.0	96.5	89.7–98.9	92.8	81.7–97.3	89.4	60.2–97.6	92.1	82.7–96.5
T3–T4	87.9	81.9–92.1	71.5	62.2–78.9	81.6	76.6–85.7	74.8	66.5–81.3	49.8	39.4–59.3	65.2	58.8–70.9
N												
N0	96.7	90.6–98.9	95.0	83.4–98.5	96.2	91.5–98.3	95.9	84.7–98.9	82.5	68.2–90.8	91.8	84.8–95.7
N1, N2, N+	85.4	76.4–91.2	63.2	47.6–75.3	78.5	70.7–84.5	61.5	50.2–70.9	32.1	19.2–45.8	52.4	43.4–60.5

**Table 4 cancers-16-02390-t004:** Reggio Emilia Cancer Registry, years 2017–2018. Cox Regression analysis, adjusted for age, stage, site, and surgery (yes/no).

Characteristics	Univariable Analysis		Multivariable Analysis by MDT		Multivariable Analysis by Year	
	HR	95% CI	HR	95% CI	HR	95% CI
MDT						
Yes	1.0	Ref.	1.0	Ref.		
No	2.6	2.0–3.3	2.0	1.5–2.6		
Year						
2017	1.0	Ref.			1.0	Ref.
2018	1.2	0.9–1.5			1.2	0.9–1.6

## Data Availability

The data presented in this study are available on request from the corresponding author. The data are not publicly available due to ethical and privacy issues; requests for data must be approved by the Ethics Committee after the presentation of a study protocol.

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
