# Peer review of "Characteristics and Outcomes of Colorectal Cancer Patients Cared for by the Multidisciplinary Team in the Reggio Emilia Province, Italy"

_cancers, 2024, doi:10.3390/cancers16132390_

Round 1

Reviewer 1 Report (Previous Reviewer 3)

Comments and Suggestions for Authors

I think that authors fully addressed my comments

Author Response

Comments and Suggestions for Authors

I think that authors fully addressed my comments

RE: We sincerely appreciate the reviewer's positive feedback and valuable suggestions, which have significantly contributed to enhancing the quality of our work.

Reviewer 2 Report (New Reviewer)

Comments and Suggestions for Authors

The study demonstrates that the overall survival is much higher in patients of colorectal cancer who was discussed in a multidisciplinary team.

1. Figure 1 needs to be revised to have detailed information of analysis time. The explanation is needed how 3 year survival was calculated in 2017-2018 data (for 2 years). 

2. Materials and Methods need to be improved to have subsection title with detailed information and numbers.

3. Results should be revised to have each subsection with the content of the result, not only Figures and Tables. Please re-organize the result section with titles of the subsections.

4. Table 4 needs to be improved to have detailed information of surgery.

Comments on the Quality of English Language

References need to be formatted according to the journal standard.

Careful proofreading is needed.

Author Response

The study demonstrates that the overall survival is much higher in patients of colorectal cancer who was discussed in a multidisciplinary team.

  1. Figure 1 needs to be revised to have detailed information of analysis time. The explanation is needed how 3 year survival was calculated in 2017-2018 data (for 2 years). 

RE: We thank the reviewer for the suggestion. We appreciate the opportunity to clarify and enhance Figure 1. The 2017-2018 cohort in our study underwent a 3-year follow-up period post-diagnosis. We have now provided detailed analyses for both individual years and the overall cohort.

  1. Materials and Methods need to be improved to have subsection title with detailed information and numbers.

RE: We appreciate the reviewer's suggestion to enhance the clarity and structure of the Materials and Methods section. In response to the comment, we have now subdivided the Materials and Methods into clear subsections, incorporating detailed information and numerical data to improve comprehensibility and facilitate easier navigation for readers.

  1. Results should be revised to have each subsection with the content of the result, not only Figures and Tables. Please re-organize the result section with titles of the subsections.

RE: As suggested, we have revised the Results section to include subsections with descriptive titles for each relevant aspect of the study findings, beyond just Figures and Tables. This reorganization aims to improve clarity, highlight key findings, and enhance the overall readability of the results for readers.

  1. Table 4 needs to be improved to have detailed information of surgery.

RE: While we appreciate the reviewer’s suggestion, we regret to inform the reviewer that detailed information on the surgery is unavailable for inclusion in Table 4. This limitation has been duly acknowledged in both the Materials and Methods section and the study's limitations section to maintain transparency regarding the available data for analysis.

Comments on the Quality of English Language

References need to be formatted according to the journal standard.

RE: Thank you for your request. We have revised the references to adhere to the journal's formatting standards.

Careful proofreading is needed.

RE: Thank you for the feedback. We have thoroughly reviewed the entire text, conducted additional linguistic revisions, and will make final corrections upon receipt of the proof.

Reviewer 3 Report (New Reviewer)

Comments and Suggestions for Authors

This paper by Mangone et al. describes the findings from the investigation of the association between multidisciplinary team (MDT) and the prognosis, by measuring relative survival rates in patients with colorectal cancer (CRC) in the Reggio Emilia province, Italy. In a multivariate regression analysis, the findings indicated an increased risk for no-MDT patients. Revisions are good. However, please add certain limitations to the study, if present.

Author Response

This paper by Mangone et al. describes the findings from the investigation of the association between multidisciplinary team (MDT) and the prognosis, by measuring relative survival rates in patients with colorectal cancer (CRC) in the Reggio Emilia province, Italy. In a multivariate regression analysis, the findings indicated an increased risk for no-MDT patients. Revisions are good. However, please add certain limitations to the study, if present.

RE: As requested, we have now incorporated additional limitations of the study at the end of the discussion section, as suggested.

Round 2

Reviewer 2 Report (New Reviewer)

Comments and Suggestions for Authors

The manuscript has been improved by the revision.

It still seems to be more comprehensive if the result section is sub-sectioned with the newly added titles with figures and tables along with the text.

This manuscript is a resubmission of an earlier submission. The following is a list of the peer review reports and author responses from that submission.

Round 1

Reviewer 1 Report

Comments and Suggestions for Authors

The manuscript entitled ‘Characteristics and outcomes of colorectal cancer patients cared 2 for by the multidisciplinary team in the Reggio Emilia prov-3 ince, Italy’ provided an interesting topic to recommend MDT for CRC patients. However, the conclusion for this manuscript is not solid based on current results.

1.       As we know, MDT usually refers to a clinical diagnosis and treatment model in which experts from two or more related disciplines form a relatively fixed expert group, through regular, timed and addressed meetings. It is a standardized and world-leading diagnosis and treatment model. So it is supposed to provide more specific therapeutic strategy for patients with a promising outcome. However, in the current manuscript, the main issue in this manuscript is the bias for different groups. In Table 2, the distribution of CRC patients was different between MDT and non-MDT groups regarding methods of diagnosis, stage and whether surgery. Although no statistical significance was found regarding age at diagnosis, the difference was significant if divided into two groups under 70 vs 70+, with p = 0.0487. All of these results indicate the bias of patients selected between different group.

2.       It is not suitable to analyze the survival information of patients with bias. It is suggested to describe the manuscript in narrative form, rather than a research article.

Comments on the Quality of English Language

The manuscript entitled ‘Characteristics and outcomes of colorectal cancer patients cared 2 for by the multidisciplinary team in the Reggio Emilia prov-3 ince, Italy’ provided an interesting topic to recommend MDT for CRC patients. However, the conclusion for this manuscript is not solid based on current results.

1.       As we know, MDT usually refers to a clinical diagnosis and treatment model in which experts from two or more related disciplines form a relatively fixed expert group, through regular, timed and addressed meetings. It is a standardized and world-leading diagnosis and treatment model. So it is supposed to provide more specific therapeutic strategy for patients with a promising outcome. However, in the current manuscript, the main issue in this manuscript is the bias for different groups. In Table 2, the distribution of CRC patients was different between MDT and non-MDT groups regarding methods of diagnosis, stage and whether surgery. Although no statistical significance was found regarding age at diagnosis, the difference was significant if divided into two groups under 70 vs 70+, with p = 0.0487. All of these results indicate the bias of patients selected between different group.

2.       It is not suitable to analyze the survival information of patients with bias. It is suggested to describe the manuscript in narrative form, rather than a research article.

Reviewer 2 Report

Comments and Suggestions for Authors

I thank the editors for the opportunity to review the manuscript by Mangone et al. The authors present an analysis on the characteristics and outcomes of patients with colorectal cancer who were either treated by a multidisciplinary care team (MDT) or not. They show improved survival rates for patients that were cared for by MDTs. In the present form, this manuscript is not suitable for publication in Cancers, as I have the following major concerns:

- Most importantly, though the authors show acceptable survival rates of MDT patients which are comparable to those in the literature, the survival rates of the other patient cohort are disastrous. In 2023, a 1-year survival rate of 62% across all stages for Italian patients needs thorough explanation. Normally, the 5-year survival rate for patients with metastases may reach equal numbers nowadays. However, I find the discussion to be insufficient. The authors need to present convincing reasons for these abysmal numbers. Do Italian doctors need to follow cancer treatment guidelines? Are case discussions in tumor boards mandatory?

- Due to the reported numbers, the treatment needs to be presented in more detail. In the present version, only "surgery yes/no" and "chemotherapy yes/no" is distinguished. Was radiotherapy administered? Which chemotherapy agents were given? What about metastases, did patients with metastases undergo surgery? 

- The presentation of results needs major improvements. For example, tables 1-3 could be summarized. The axes of the Kaplan Meier graphs are not labelled. In Table 1, the first rows are missing.

Comments on the Quality of English Language

Language is fine.

Reviewer 3 Report

Comments and Suggestions for Authors

Mangone et al. investigated the association between multidisciplinary team (MDT) and the prognosis, as measured by relative survival rates in colorectal cancer patients. In a multivariate regression analysis an increased risk for no-MDT patients [OR 2.3; 95% CI 1.8-2.9) was shown.

The study is very important and good; there are three issues:

1.      Authors should describe in the introduction what specialties a MDT is consisting of.  “Multidisciplinary” means different disciplines. Which different disciplines are included in MDT along with oncologists?

2.      Authors should re-phrase some result descriptions, as they always write about “an increased risk for no-MDT patients”. But having retrospective design, it is not usual to conclude about risk, but only about associations.  When authors would write “no-MDT was associated with increased risk of XXXX” or better “Being treated by MDT was associated with increased survival time”, this would be correctly.

3.      Authors should not use terms like “multivariate analysis”, bur better “multivariable regression analysis”.  There different kind of multivariate analyses, and authors used ‘multivariable logistic regression analysis”.